# Selective Inhibition of Cardiac Late Na^+^ Current Is Based on Fast Offset Kinetics of the Inhibitor

**DOI:** 10.3390/biomedicines11092383

**Published:** 2023-08-25

**Authors:** Muhammad Naveed, Aiman Saleh A. Mohammed, Leila Topal, Zsigmond Máté Kovács, Csaba Dienes, József Ovári, Norbert Szentandrássy, János Magyar, Tamás Bányász, János Prorok, Norbert Jost, László Virág, István Baczkó, András Varró, Péter P. Nánási, Balázs Horváth

**Affiliations:** 1Department of Pharmacology and Pharmacotherapy, Faculty of Medicine, University of Szeged, H-6720 Szeged, Hungary; muhammadnaveedkhan01@gmail.com (M.N.); mohammed.aiman@med.u-szeged.hu (A.S.A.M.); topal.leila@gmail.com (L.T.); jost.norbert@med.u-szeged.hu (N.J.); virag.laszlo@med.u-szeged.hu (L.V.); baczko.istvan@med.u-szeged.hu (I.B.); varro.andras@med.u-szeged.hu (A.V.); 2Department of Physiology, Faculty of Medicine, University of Debrecen, H-6720 Debrecen, Hungary; kovacs.zsigmond@med.unideb.hu (Z.M.K.); dienes.csaba@med.unideb.hu (C.D.); ovari.jozsef@med.unideb.hu (J.O.); szentandrassy.norbert@med.unideb.hu (N.S.); magyar.janos@med.unideb.hu (J.M.); banyasz.tamas@med.unideb.hu (T.B.); horvath.balazs@med.unideb.hu (B.H.); 3Department of Basic Medical Sciences, Faculty of Dentistry, University of Debrecen, H-6720 Debrecen, Hungary; 4Division of Sport Physiology, Department of Physiology, Faculty of Medicine, University of Debrecen, H-6720 Debrecen, Hungary; 5ELKH-SZTE Research Group for Cardiovascular Pharmacology, Loránd Eötvös Research Network, 1097 Szeged, Hungary; prorok.janos@med.u-szeged.hu; 6Division of Dental Physiology and Pharmacology, Faculty of Dentistry, University of Debrecen, H-6720 Debrecen, Hungary

**Keywords:** late Na^+^ current, class I antiarrhythmics, rate-dependent block, dog myocytes, GS967

## Abstract

The present study was designed to test the hypothesis that the selectivity of blocking the late Na^+^ current (I_NaL_) over the peak Na^+^ current (I_NaP_) is related to the fast offset kinetics of the Na^+^ channel inhibitor. Therefore, the effects of 1 µM GS967 (I_NaL_ inhibitor), 20 µM mexiletine (I/B antiarrhythmic) and 10 µM quinidine (I/A antiarrhythmic) on I_NaL_ and I_NaP_ were compared in canine ventricular myocardium. I_NaP_ was estimated as the maximum velocity of action potential upstroke (V^+^_max_). Equal amounts of I_NaL_ were dissected by the applied drug concentrations under APVC conditions. The inhibition of I_NaL_ by mexiletine and quinidine was comparable under a conventional voltage clamp, while both were smaller than the inhibitory effect of GS967. Under steady-state conditions, the V^+^_max_ block at the physiological cycle length of 700 ms was 2.3% for GS967, 11.4% for mexiletine and 26.2% for quinidine. The respective offset time constants were 110 ± 6 ms, 456 ± 284 ms and 7.2 ± 0.9 s. These results reveal an inverse relationship between the offset time constant and the selectivity of I_NaL_ over I_NaP_ inhibition without any influence of the onset rate constant. It is concluded that the selective inhibition of I_NaL_ over I_NaP_ is related to the fast offset kinetics of the Na^+^ channel inhibitor.

## 1. Introduction

Following the inactivation of the peak Na^+^ current (I_NaP_), responsible for the action potential upstroke, a sustained Na^+^ current component with a much lower density referred to as the late Na^+^ current (I_NaL_) remains active during the entire plateau of the cardiac action potential (AP). I_NaL_ is believed to represent approximately half of the Na^+^ entry into cardiomyocytes and thus may intimately influence the Na^+^ and, consequently, the Ca^2+^ homeostasis of cardiac cells. The elevation of I_NaL_ causes an inward shift between the finely tuned balance of inward and outward currents and results in Ca^2+^ overload—both changes are known to be proarrhythmic increasing the risk of sudden cardiac death. Indeed, an increased I_NaL_ has been reported under several pathological conditions including heart failure [1,2,3], hypertrophic cardiomyopathy [4] or LQT3 [5], all associated with a higher arrhythmia propensity [1,6,7,8]. Conventionally, these arrhythmias were treated using class I antiarrhythmic agents, since several class I drugs were shown to inhibit I_NaL_ in addition to the suppression of I_NaP_. However, these drugs also decrease the maximum rate of depolarization during the AP upstroke (V^+^_max_), an effect which may increase the risk of cardiac arrhythmias [9,10,11,12,13,14,15]. As a result of this, agents suppressing I_NaL_ selectively were developed in the past decades [2,16,17,18,19]. One of these agents, GS967, was a particularly selective candidate when tested in rabbit ventricular myocytes [16]. While considerable interspecies differences have been observed in the electrophysiological properties of various mammalian cardiac preparations [20,21], canine ventricular myocytes are considered a reasonably good model for human ventricular cells [22,23,24]. In addition, the kinetic properties of I_NaL_ were shown to be similar in dog and human [20] but different from those of other mammalian species, like pig, rabbit and guinea pig [25,26,27].

Previously, we have reported mexiletine-like effects of GS967 on I_NaL_ and V^+^_max_ (used as a surrogate indicator of I_NaP_) [28]. In the present study, therefore, we examined the inhibitory effects of GS967 on I_NaL_ and V^+^_max_, comparing them to the effects of two class I antiarrhythmic agents, the I/B drug mexiletine and the I/A compound quinidine in canine ventricular preparations. The applied drug concentrations were chosen to yield an identical inhibition of the I_NaL_ amplitude under action potential voltage clamp (APVC) conditions. Our results indicate that the selective inhibition of I_NaL_ over I_NaP_ is related to the fast offset kinetics of the Na^+^ channel inhibitor.

## 2. Materials and Methods

### 2.1. Animals

Adult mongrel dogs of either sex (n = 37) were i.m. injected with 10 mg/kg ketamine hydrochloride (Calypsol, Richter Gedeon, Hungary) plus 1 mg/kg xylazine hydrochloride (Sedaxylan, Eurovet Animal Health BV, The Netherlands) in order to achieve deep anesthesia. All experimental interventions were carried out in accordance with relevant guidelines and regulations as outlined in the ARRIVE guidelines.

### 2.2. Isolation of Cardiomyocytes

Single canine ventricular myocytes were isolated using a segment perfusion technique, as previously described [29]. Briefly, the LAD coronary artery was cannulated and used for supplying solutions required for the dissociation of the individual cells with minimal injury. After dissection, the tissue was first perfused with a nominally Ca^2+^-free solution (Minimum Essential Medium Eagle, Joklik Modification, Sigma-Aldrich Co., St. Louis, MO, USA) for 5 min in order to reduce the Ca^2+^ load and also to promote the uncoupling of cell-to-cell junctions. Following this, the perfusate was supplemented with 1 mg/mL collagenase (Type II, final activity of 224 U/mL, Worthington Biochemical Co., Lakewood, NJ, USA) and 0.2% bovine serum albumin (Fraction V., Sigma) in the presence of 50 µM Ca^2+^. This step was applied for 30 min to digest the collagen fibers in the extracellular matrix, resulting in the release of individual cells. Finally, the normal Ca^2+^ concentration was gradually restored in the bathing medium. The cells were kept at 15 °C before use, allowing normalization of their intracellular ionic composition. All chemicals applied in these experiments were purchased from Sigma-Aldrich Co. (St. Louis, MO, USA).

### 2.3. Electrophysiology

The cells, studied by an inverted microscope, were continuously superfused with a modified Tyrode at a rate of 1–2 mL/min. The modified Tyrode solution contained (in mM): NaCl: 121; KCl: 4; CaCl_2_: 1.3; MgCl_2_: 1; HEPES: 10; NaHCO_3_: 25; glucose: 10; pH: 7.35; osmolarity: 300 ± 3 mOsm. The temperature of this perfusate was set to 37 °C with a temperature controller (Cell MicroControls, Norfolk, VA, USA). Electrical signals were generated, amplified and recorded using a MultiClamp 700A or 700B amplifier under the control of pClamp 10 software (Molecular Devices, Sunnyvale, CA, USA) after analogue-to-digital conversion (Digidata 1440A or 1332, Molecular Devices). The electrodes were manufactured from borosilicate glass. The tip resistance of these micropipettes ranged between 2 and 3 MΩ when filled with pipette solution. Ion currents were recorded under whole-cell voltage clamp conditions. The series resistance was typically 4–8 MΩ. The measurement was discarded when the series resistance changed substantially during the experimental period.

#### 2.3.1. Action Potential Voltage Clamp

APVC experiments were performed according to the methods described previously [30,31]. This technique allows for the visualization of various ion currents flowing during an actual ventricular AP by using a selective inhibitor of the respective ion current. In these experiments, therefore, a midmyocardial canine ventricular AP was applied as a command signal. The current traces were continuously recorded before and after a 5 min superfusion with the applied Na^+^ channel inhibitor. The drug-sensitive current was obtained by subtracting the post-drug trace from the reference pre-drug trace. When measuring I_NaL_, the external solution was supplemented with 1 µM nisoldipine, 1 µM E4031 and 100 µM chromanol 293B, in order to eliminate contaminations by Ca^2+^ and K^+^ currents. The pipette solution contained (in mM): K-aspartate, 120; KCl, 30; MgATP, 3; HEPES, 10; Na_2_-phosphocreatine, 3; EGTA, 0.01; cAMP, 0.002; KOH, 10; at pH = 7.3; osmolarity of 285 mOsm. The amplitude of I_NaL_ was measured at 50% duration of the APD_90_ value of the command AP. The initial 20 ms period following the AP upstroke was excluded from evaluation when determining the current integral in order to eliminate the contribution of I_NaP_. A total of 20 consecutive current traces were averaged and analyzed in each experiment to reduce the noise and the trace-to-trace fluctuations of the AP configuration. Ion currents were normalized to cell capacitance, determined by applying a short (15 ms) hyperpolarizing pulse from +10 mV to −10 mV.

#### 2.3.2. Conventional Voltage Clamp

Rectangular command pulses were applied in conventional voltage clamp experiments to study the effects on I_NaL_ at stable test potentials, as described previously [28]. The cells were perfused with HEPES-buffered Tyrode solution containing (in mM): NaCl, 144; NaH_2_PO_4_, 0.4; KCl, 4.0; CaCl_2_, 1.8; MgSO_4_, 0.53; glucose, 5.5; HEPES, 5.0; at pH = 7.4. This solution contained 1 µM nisoldipine, 0.5 µM HMR-1556 and 0.1 µM dofetilide in order to block the Ca^2+^ and K^+^ currents. The pipette solution contained (in mM): CsCl, 125; TEACl, 20; MgATP, 5; EGTA, 10; HEPES, 10; at pH = 7.2. Test pulses of 2 s duration were clamped to −20 mV from the holding potential of −120 mV. The total amount of I_NaL_ was determined by pharmacological subtraction using a final superfusion with 20 µM TTX. The amplitude of I_NaL_ was determined at 50 ms following the start of the pulse. When measuring the charge carried by I_NaL_ (integral), the initial 20 ms of the current was excluded from evaluation to minimize the contamination with I_NaP_.

#### 2.3.3. Recording of APs from Multicellular Ventricular Preparations

Right ventricular papillary muscles were applied for AP recordings using the conventional sharp microelectrode technique, as described previously [14,28]. Membrane potential changes were recorded at 37 °C using sharp microelectrodes filled with 3 M KCl (tip resistance 10–20 MΩ) and connected to the input of an amplifier (MDE GmbH, Heidelberg, Germany). The preparations were stimulated with current pulses having durations of 1 ms and amplitudes of 200% diastolic threshold using a pair of platinum electrodes. Before starting the experiment, the preparations were paced at a cycle length of 1 s for at least 60 min allowing equilibration. After this period, the cycle length was sequentially changed between 0.3 and 5 s and the 25th AP was recorded at each cycle length. A quasi steady-state rate-dependence could rapidly be obtained under these conditions. APs were sampled at 100 kHz with an ADA 3300 data acquisition board (Real Time Devices Inc., State Collage, PA, USA) and stored for later analysis. After taking control records, the preparations were treated with 1 µM GS967, 20 µM mexiletine or 10 µM quinidine for 20 min, then the protocol was repeated.

#### 2.3.4. Determination of Offset Kinetics

The restitution kinetics of V^+^_max_ were analyzed to determine the offset time constant. Initially, the preparations were paced at a constant cycle length of 1 s (basic cycle length). Each train, containing 20 basic stimuli, was followed by a single extra stimulus delivered with a successively increasing coupling interval. Accordingly, each 20th basic AP was followed by a single extra AP appearing after gradually increasing diastolic intervals (defined as the time elapsed from the APD_90_ of the last basic AP to the upstroke of the extra AP). The restitution curves were obtained by plotting the V^+^_max_ of each extra AP against the respective diastolic interval. Data were fitted to a single exponential function.

#### 2.3.5. Determination of Onset Kinetics

The onset kinetics of drug action on V^+^_max_ were determined by stimulating the preparation at a short cycle length of 0.4 s following a 1 min period of rest and the initial 40 APs were recorded. The V^+^_max_ values of the consecutive APs were plotted as a function of the number of the AP within the train. The onset rate constant was determined by fitting the consecutive V^+^_max_ values to a monoexponential function.

### 2.4. Statistics

Results are given as mean ± SEM values; n indicates the number of preparations studied. Statistical significance was evaluated using a one-way ANOVA followed by Student’s *t*-test for paired or unpaired data, or Tukey’s post-hoc test where pertinent. Differences were considered significant when *p* < 0.05. Microsoft Excel software was applied for statistical analysis, and curve fitting was performed using Origin 2021b software.

## 3. Results

### 3.1. Effects of GS967, Mexiletine and Quinidine under APVC Conditions

The effects of 1 µM GS967, 20 µM mexiletine and 10 µM quinidine on I_NaL_ were studied under APVC conditions using canonic mid-myocardial APs as command signals. Each of the studied drugs dissected similar inward current profiles corresponding to I_NaL_ (Figure 1A). Furthermore, there were no significant differences in either the current densities measured at 50% duration of APD_90_ of the command AP (−0.37 ± 0.07 A/F for GS967, −0.375 ± 0.07 A/F for mexiletine and −0.377 ± 0.06 A/F for quinidine), or in the current integrals (−56.7 ± 9.1 mC/F for GS967, −65.2 ± 11.5 mC/F for mexiletine and −61.2 ± 8.8 mC/F for quinidine), as shown in Figure 1B and Figure 1C, respectively.

### 3.2. Effects of GS967, Mexiletine and Quinidine on I_NaL_ under Conventional Voltage Clamp Conditions

The effects of 1 µM GS967, 20 µM mexiletine and 10 µM quinidine on I_NaL_ were also evaluated using a conventional voltage clamp. In these experiments, the membrane potential was clamped to −20 mV from the holding potential of −120 mV for 2 s (Figure 2).

The density of I_NaL_ was determined at 50 ms after the beginning of the pulse. Each drug decreased the I_NaL_ density significantly (*p* < 0.05): from −0.313 ± 0.035 to −0.062 ± 0.01 A/F with 1 µM GS967, from −0.337 ± 0.035 to −0.197 ± 0.026 A/F with 20 µM mexiletine and from −0.333 ± 0.043 to −0.186 ± 0.02 A/F with 10 µM quinidine. This suppression of I_NaL_ density corresponds to 80.4 ± 2.2% (n = 6), 41.9 ± 3.7% (n = 5) and 43.3 ± 2.2% (n = 6) inhibition, respectively. Comparing the charge carried by the current gave similar results: the current integrals were reduced significantly from −69.1 ± 7.9 to −15.4 ± 3.9 mC/F with 1 µM GS967, from −68.6 ± 6.2 to −37.0 ± 4.7 mC/F with 20 µM mexiletine and from −68.2 ± 11.7 to −42.9 ± 9.2 mC/F with 10 µM quinidine (*p* < 0.05 each), corresponding to 79.0%, 46.1% and 37.1% inhibition, respectively.

### 3.3. Effects of GS967, Mexiletine and Quinidine on Action Potential Upstroke

To compare the effects of GS967, mexiletine and quinidine on I_NaP_, the maximum rate of depolarization (V^+^_max_) was determined during the AP upstroke. V^+^_max_ was used as an indicator of I_NaP_ [32] in order to avoid the technical difficulties related to the direct measurement of I_NaP_ at 37 °C, bearing in mind that V^+^_max_ is only an approximate but not a linear measure of I_NaP_ [33,34]. These experiments were performed in multicellular ventricular preparations (right ventricular papillary muscles) using sharp microelectrodes, an arrangement which resulted in a high stability of V^+^_max_ measurements. Under steady-state conditions, V^+^_max_ was significantly reduced in the entire frequency range up to the longest cycle length of 5 s by 10 µM quinidine, while this effect with 20 µM mexiletine and 1 µM GS967 was significant only at cycle lengths shorter than 700 and 500 ms, respectively (Figure 3). At the physiological cycle length of 700 ms, the magnitude of V^+^_max_ block was: 2.3% with GS967, 11.4% with mexiletine and 26.2% with quinidine.

The offset kinetics of the Na^+^ channel inhibitors were studied using the protocol described in the methods. Accordingly, trains of steady cycle length were followed by extra stimuli, applied with gradually increasing coupling intervals. The time constant of recovery of V^+^_max_ was determined by applying extra stimuli with gradually increasing diastolic intervals following a constant frequency stimulation at 1 Hz. Each V^+^_max_ value was plotted against the respective coupling interval and the obtained restitution curves were fitted to a monoexponential function to determine the offset time constants. As shown in Figure 4A–C, the offset time constant was 110 ± 6 ms for GS967, 456 ± 284 ms for mexiletine and 7.2 ± 0.9 s for quinidine. The onset kinetics of the V^+^_max_ block were studied during a train of constant stimulation, delivered at a rate of 2.5 Hz, applied after a 1 min stimulation-free period. In this case, the initial 40 APs were analyzed. The onset rate constant was 6.4 ± 0.9 AP for 1 µM GS967, 3.4 ± 0.4 AP for 20 µM mexiletine and 5.6 ± 0.4 AP for 10 µM quinidine (Figure 4D–F).

### 3.4. Selectivity of I_NaL_ over I_NaP_ Inhibition

Using the results obtained in the experiments above, the selectivity of inhibition of I_NaL_ over I_NaP_ can be estimated. The percentage of the V^+^_max_ block, obtained at the physiological cycle length of 700 ms under steady-state conditions, was divided by the percentage of I_NaL_ inhibition measured under conventional voltage clamp conditions. The results are presented in the ordinate of Figure 5A and plotted against the offset time constant of the inhibitor.

According to this type of visualization, lower values in the ordinate represent a higher selectivity. It is clearly shown in the figure that the selectivities of the studied Na^+^ channel inhibitors on I_NaL_ over I_NaP_ are inversely proportional with the offset time constant of the agent studied. In other words, faster offset kinetics are associated with a higher I_NaL_ selectivity. Identical results were obtained when the I_NaL_ current densities and integrals, dissected by GS967, mexiletine and quinidine under APVC conditions, were analyzed in a similar way. These results are demonstrated in Figure 5B, where the absolute magnitude of the V^+^_max_ block was divided by the absolute magnitude of the dissected mid-plateau current density or integral. Although not shown, there was no relationship between these selectivity ratios and the offset rate constants of the inhibitors.

## 4. Discussion

In this study the effects of the selective I_NaL_-inhibitor GS967 were compared to those of the class I/B antiarrhythmic drug mexiletine and the I/B agent quinidine on I_NaL_ and I_NaP_ in canine ventricular myocardium by combining the conventional voltage clamp, APVC and sharp microelectrode techniques, where I_NaP_ was represented by V^+^_max_ during the AP upstroke [32]. While the applied drug concentrations revealed a similar I_NaL_ under APVC conditions, their suppressive effect on V^+^_max_ was largely different, i.e., it was more and more pronounced with the increasing offset time constant of the drug used. Importantly, the pacing cycle length was equally 700 ms when the results of steady-state V^+^_max_ measurements were compared to those obtained for I_NaL_ under APVC conditions. Similar conclusions can be drawn from those experiments when I_NaL_ was determined using a conventional voltage clamp. In this case—although 1 µM GS967 caused a larger inhibition of I_NaL_ than 20 µM mexiletine or 10 µM quinidine—the GS967-induced V^+^_max_ block was weaker than observed with mexiletine and especially with quinidine. Under steady-state conditions, V^+^_max_ was reduced significantly only at the shortest cycle lengths of 300 and 400 ms by GS967 (offset time constant = 110 ms), and between 300 and 500 ms cycle lengths by mexiletine (offset time constant = 456 ms), while it was suppressed in the full range of the cycle lengths from 300 ms to 5 s by quinidine (offset time constant = 7.2 s). The selectivity of the drugs on I_NaL_ over I_NaP_ inhibition was characterized by the ratio of the V^+^_max_ block and I_NaL_ inhibition, where a lower value means a higher selectivity. The results summarized in Figure 5 clearly demonstrate the inverse relationship between the offset time constant and the I_NaL_ over I_NaP_ selectivity—independently of the method of the current measurement and considering the current density or current integral.

Considering the drug–channel interaction in line with the modulated [35] or guarded [36] receptor theory, drug binding to the Na^+^ channel occurs during the AP upstroke, when the membrane gets depolarized. After repolarization, the drug may leave the binding site, resulting in recovery from the V^+^_max_ block after a certain period of rest. At the physiological cycle length of 700 ms, this recovery may almost be fully complete in the case of GS967, and intermediate with mexiletine, while only negligible with quinidine. In contrast to I_NaP_, which is the result of one single opening of the Na^+^ channel, there are prolonged (“burst mode”) or repetitive (“late scattered mode”) openings underlying I_NaL_ [37]. Therefore, in the case of I_NaL_, the offset kinetics may have a limited influence on the extent of the block due to the possibility of repetitive drug (re)binding to the channel. A few years ago, GS967 was categorized as novel class VI antiarrhythmic agent based on its selective blockade on I_NaL_ [38]. Our results suggest that this selectivity is closely related to fast offset kinetics of GS967, consequently, such a classification may be questioned. Instead, GS967 should be considered as a class I/B agent with extremely fast offset kinetics.

Cardiac Na^+^ current is partly mediated by Na^+^ channels different from the cardiac Na_v_1.5 channel, such as Na_v_1.8 [39,40]. Similarly, a relatively high level of Na_v_2.1 mRNA expression was observed in human ventricular myocardium [41], although evidence for its actual functional contribution is still missing. If these non-cardiac Na^+^ channels contribute to I_NaL_ but not I_NaP_, their pharmacological blockade might actually cause selective I_NaL_ inhibition. GS967 does not inhibit these non-cardiac Na^+^ channel isoforms (Na_v_1.8, Na_v_2.1). Although the neuronal Na_v_1.1 was sensitive to GS967 [42], its contribution to cardiac I_NaL_ has not been demonstrated. In contrast, GS967 has been shown to strongly block the cardiac Na_v_1.5 channel isoform [9,10].

Na^+^ channels have multiple open and closed channel states with different drug binding affinities to their open, inactivated or resting closed states [37]. Therefore, drugs interacting differently with the binding sites depending on their actual open or closed states may result in different inhibitory effects on I_NaL_ and I_NaP_. When a blocker binds rapidly and with a high affinity to an open or inactivated state, and it dissociates rapidly from the closed resting state, it will inhibit I_NaL_ but not I_NaP_ because of the complete drug dissociation in the resting closed state if the stimulation frequency is not too high, i.e., when the pacing cycle length is much longer than the offset time constant of the drug. As a consequence, whether a drug apparently inhibits I_NaP_ or I_NaL_ selectively largely depends on the stimulation protocol. Based on our present results and other recent data [9,10], the mechanism of “specific I_NaL_ inhibition” in contrast to the inhibition of I_NaP_ seems to depend only on the applied pacing frequency and the kinetics of the drug–channel interaction. Accordingly, the effect of GS967 is not fundamentally different from other class I/B antiarrhythmic drugs, like mexiletine [14,37], lidocaine [9,10,14], amiodarone [11] and ranolazine [43], except for its extremely fast offset kinetics. This interpretation is in accordance with the high (38-fold) selectivity of ranolazine [2], the intermediate (13-fold) selectivity of amiodarone [3], and the low (3-fold) selectivity of flecainide [44] reported for I_NaL_ over I_NaP_.

Taken together, the present results and the literature suggests that GS967 and similar compounds should be classified as potent class I/B antiarrhythmic agents with ultra-rapid offset kinetics [45]. The results also suggest that investigations of “selective” I_NaL_ inhibitors should be carried out through a wide range of stimulation frequencies, since the effect of drugs with fast offset kinetics for I_NaP_ inhibition can easily be misinterpreted.

## Figures and Tables

**Figure 1 biomedicines-11-02383-f001:**
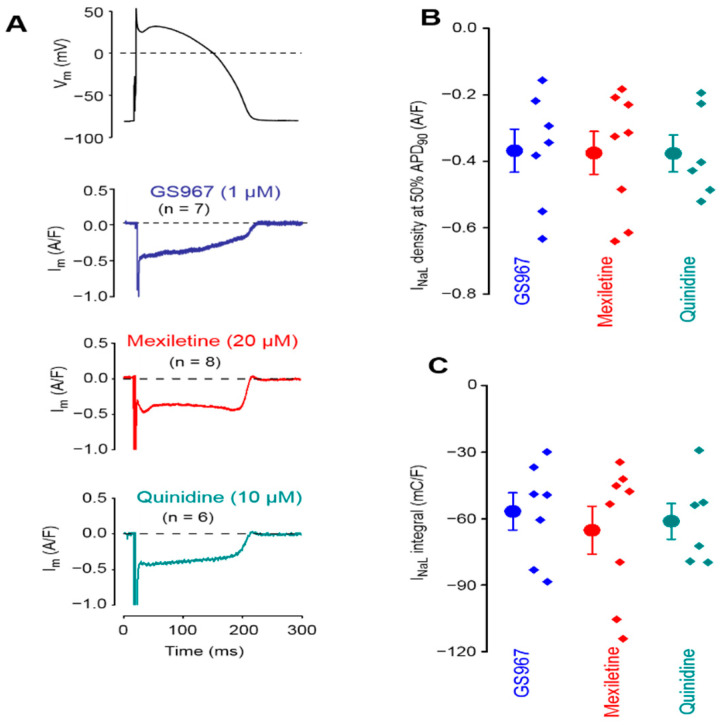
Averaged analogue current records exhibited by 1 µM GS967, 20 µM mexiletine and 10 µM quinidine under action potential voltage clamp conditions in isolated canine ventricular myocytes (**A**). The stimulation cycle length was 700 ms. The command AP is shown above the current traces. Dashed lines indicate zero voltage and current levels. (**B**) Current densities measured at 50% duration of APD_90_ of the command AP. (**C**) Current integrals from which the initial 20 ms period was excluded. Symbols and bars are the mean ± SEM, small dots represent individual data, and numbers in parentheses indicate the number of myocytes studied.

**Figure 2 biomedicines-11-02383-f002:**
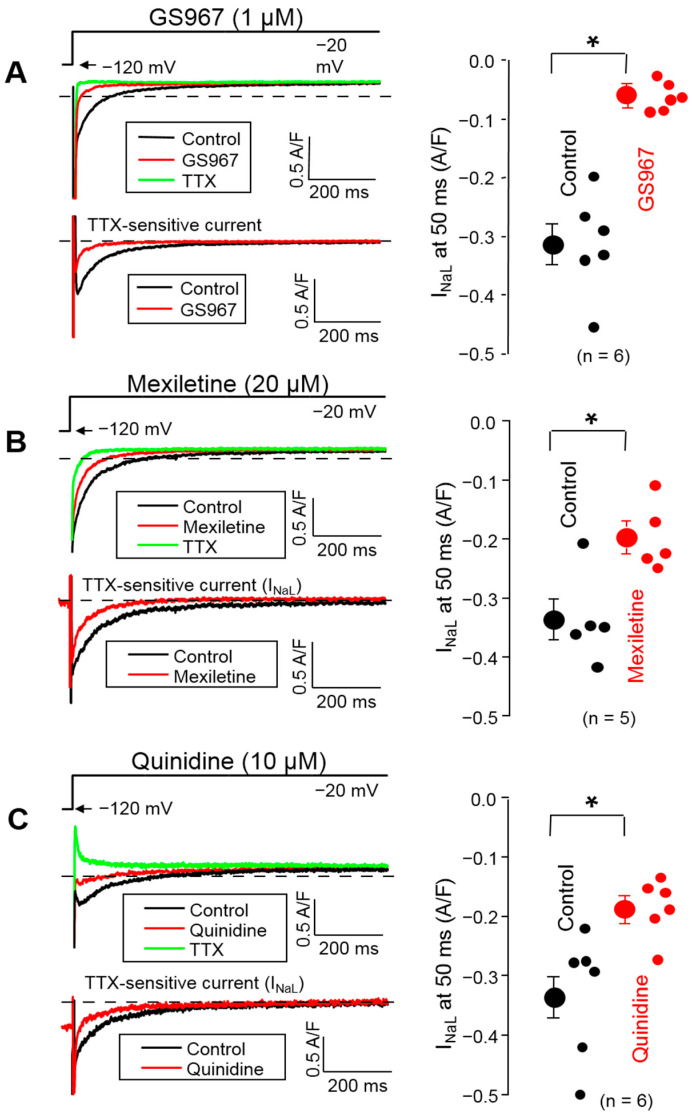
Representative superimposed analogue records (left) and average I_NaL_ densities, measured at 50 ms, (right) demonstrating the effects of 1 µM GS967 (**A**), 20 µM mexiletine (**B**) and 10 µM quinidine (**C**) on I_NaL_ under conventional voltage clamp conditions using test pulses of 2 s duration clamped to −20 mV from the holding potential of −120 mV. At the end of each experiment, the cells were superfused with 20 µM TTX to dissect the remaining I_NaL_. Dashed lines indicate zero current level. Symbols and bars are the mean ± SEM, small dots represent individual data, numbers in parentheses indicate the number of myocytes studied, and asterisks denote significant differences between the pre-drug and post-drug values.

**Figure 3 biomedicines-11-02383-f003:**
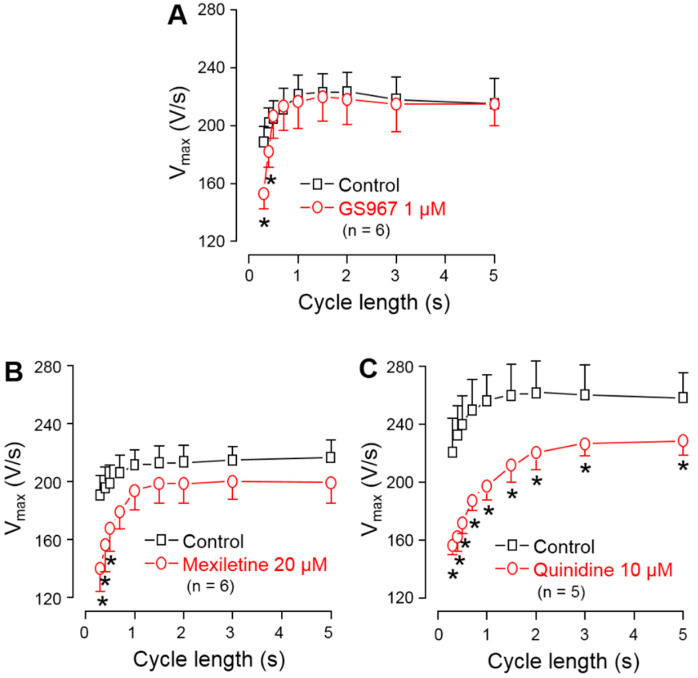
Effects of 1 µM GS967 (**A**), 20 µM mexiletine (**B**) and 10 µM quinidine (**C**) on the maximal rate of depolarization (V^+^_max_) measured in canine right ventricular trabeculae under steady-state conditions. Symbols and bars are the mean ± SEM, numbers in parentheses indicate the number of preparations studied, and asterisks denote significant differences between the pre-drug control and post-drug values.

**Figure 4 biomedicines-11-02383-f004:**
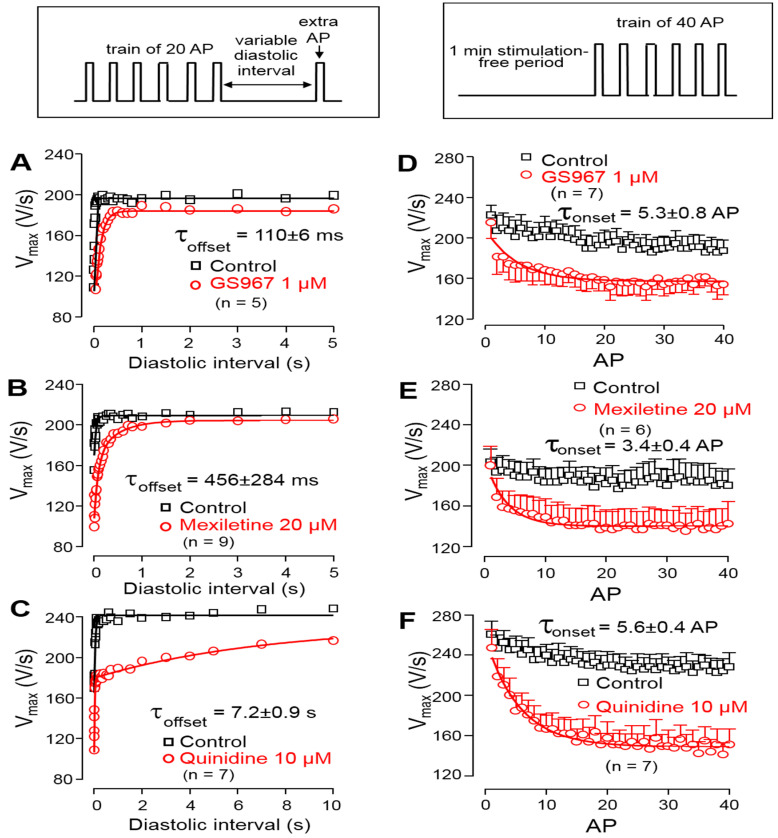
Left: Determination of the offset kinetics of GS967 (**A**), mexiletine (**B**) and quinidine (**C**) as indicated by the time-dependent restitution of V^+^_max_. Solid lines were obtained by monoexponential fitting of the data from 0 to 5 s for GS967 and mexiletine, and from 0 to 60 s for quinidine. Right: Determination of the onset kinetics of V^+^_max_ block for GS967 (**D**), mexiletine (**E**) and quinidine (**F**) using trains of 2.5 Hz applied following a 1 min stimulation-free period. Solid lines were obtained by monoexponential fit. Symbols and bars are the mean ± SEM, and numbers in parentheses indicate the number of preparations studied. The experimental protocols are shown in the insets (**top**).

**Figure 5 biomedicines-11-02383-f005:**
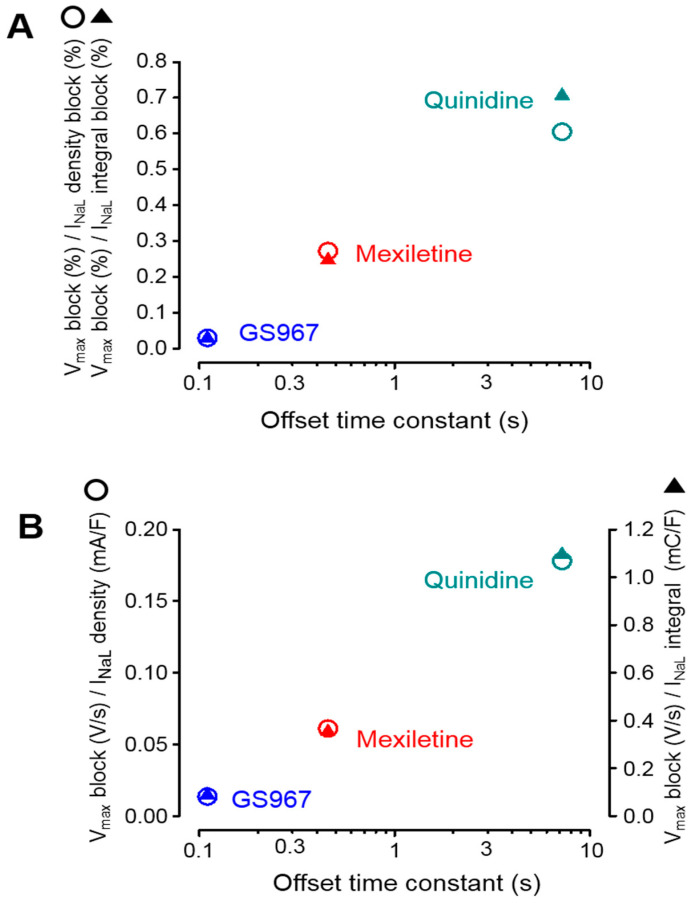
Relationship between the magnitude of the V^+^_max_ block and inhibition of I_NaL_ plotted as a function of the respective offset time constant obtained for GS967, mexiletine and quinidine, as documented in Figure 4. (**A**) Results obtained from sharp microelectrode and conventional voltage clamp experiments (Figure 2 and Figure 3), where the ratio of the percentage V^+^_max_ block was normalized to the percentage inhibition of I_NaL_ density and its integral. (**B**) Results obtained from sharp microelectrode and APVC experiments (Figure 1 and Figure 3), where the absolute magnitude of the V^+^_max_ block was divided by the density and integral of the excised I_NaL_. Open circles and filled triangles represent data regarding the current densities and integrals, respectively.

## Data Availability

The datasets generated and/or analyzed during the current study are available in the Open Science Framework repository, available online: https://osf.io/stx4v/?view_only=78cae0adb2e44f73a7fd940aa6c26a34 (https://doi.org/10.17605/OSF.IO/STX4V) (accessed on 21 August 2023).

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
