# Peer review of "Selective Inhibition of Cardiac Late Na^+^ Current Is Based on Fast Offset Kinetics of the Inhibitor"

_biomedicines, 2023, doi:10.3390/biomedicines11092383_

Round 1
Reviewer 1 Report
This manuscript is about a study on the cardiac late Na+ current on fast offset kinetics. This work is topical and thorough. I only have some minor concerns in this work:
- Title: The title has assumed the hypothesis is true. Therefore, it is better to rephrase it to “Study on …”
- Introduction: a literature review of related works and the background of this study may extend this section. The content of this section should be increased.
- Section 2.1: Please indicate the number of dogs used in this study.
- Section 2.4: Please provide the software to carry out the statistical analysis.
- L180: Why “Fig. 2” has red font?
- Figure 2 A, B and C: Why there is no data point in the subplots of A/F vs. ms?
- Please provide error bars for Figure 4 A, B and C.
Author Response
Response to Reviewer 1.
- Since the results supported our hypothesis, it seems more informative to keep the original title. The periodicals always suggest to use a title as informative as it is possible.
- As suggested by the Reviewer, a short extension was added to the introduction section.
- The total number of dogs used in the study was 37. This information was added to the revised methods section.
- Microsoft Excel software was applied for statistical analysis, curve fitting was performed using Origin 2021b software. This information was added to the revised methods section.
- It was a mistake to show “Fig.2” with red characters. It is corrected in the revised text.
- Because these are calibration bars but not subplots in Figs.2.a,b,c.
- The error bars were omitted from Figs.4.a,b,c because of the marked overlaps between the pre-drug and post-drug curves obtained in the offset kinetic studies. In contrast, the error bars are presented in the plots of Figs.4.d,e,f demonstrating the effects on onset kinetics.
The authors thank this Reviewer for his/her suggestions, which helped to further improve the manuscript.
Reviewer 2 Report
The authors studied the hypothesis that the selectivity of blocking late Na+ current over peak Na+ current is related to fast offset kinetics of the Na+ channel inhibitor. Therefore, the effects of 1μM GS967, 20μM mexiletine and 10μM quinidine on INaL and INaP were compared in canine ventricular myocardium. It was concluded that the selective inhibition of INaL over INaP is related to fast offset kinetics of the Na+ channel inhibitor.
Comments:
1.Please add how many dogs did you used in the study
2. To make the article easier to read, please detail the following statements: - Single canine ventricular myocytes were isolated using an enzymatic dispersion technique based on segment perfusion, as previously described- APVC experiments were performed according to the methods described previously
3. At discussions add please a justification for choosing dogs as an experimental model and which are the human-dog similarities.
The study is interesting, with an adequate design and the results are also well presented. In my opinion, after a minor revison the article is suitable to be publish in Biomedicine Journal.
Minor editing of english language is required
Author Response
Response to Reviewer 2.
- The total number of dogs used in the study was 37. This information was added to the revised methods section.
- Description of the experimental techniques was extended in the revised 2.2 and 2.3.1 chapters.
- This was made in the introduction section: “canine ventricular myocytes are considered a reasonably good model for human ventricular cells[22-24]. In addition, the kinetic properties of INaL were shown to be similar in dog and human[20], but different from those of other mammalian species, like pig, rabbit and guinea pig[25-27].”
In dog and human the amplitude of INa-late is monotonically decreasing during the AP, while in guinea pig and pig its amplitude is increasing during the AP plateau. The reason for this discrepancy is the difference in the slow inactivation kinetics of the INa-late current. In human and dog the slow inactivation is fast enough to close the channel before the increment of the driving force during repolarization, while in guinea pig and pig this inactivation is slower allowing to rise the current in response to the increased driving force.
The authors thank this Reviewer for his/her suggestions, which helped to further improve the manuscript.
Reviewer 3 Report
Dear authors,
Thank you for this interesting manuscript on the selective inhibition of cardiac late Na+ currents.
I am a clinician, so I can just assess it from a clinical and overall standpoint.
1) Title: This is very vague. Try to incorporate your main finding together with a bit of applicability; make it appealing to readers.
2) Introduction: This should be better structured and stay also readable to a non-electrophysiologist. Try to take a reader at their hand and guide them through, bit for bit adding more information.
3) Methods: This should also be better structured, especially with the many subheadings. There needs to be a logical path through this section.
4) Discussion: Try structuring this part with subheadings. Also add a paragraph on future applications of these findings and potential clinical applicability in the future.
The whole manuscript should be revised by an English native speaker.
Author Response
Response to Reviewer 3.
- Thank you for the suggestion. To clarify the title three words were added: “Selective inhibition of cardiac late Na+ current is based on fast offset kinetics of the inhibitor”.
- The introduction section was extended to help the non-electrophysiologist reader.
- As suggested, two more subheadings were introduced in the revised methods section: 3.4. Determination of offset kinetics, and 2.3.5. Determination of onset kinetics.
- The last sentence of discussion section contains the message for the future: “The results also suggest that investigations of “selective” INaL inhibitors should be carried out through a wide range of stimulation frequencies since the effect of drugs having fast offset kinetics for INaP inhibition, can easily be misinterpreted.”
The authors thank this Reviewer for his/her suggestions, which helped to further improve the manuscript.
Round 2
Reviewer 3 Report
Dear authors,
Thank you for having addressed my comments. As a last point, I recommend having the whole manuscript revised again by an English native speaker to improve the "flow" of reading.
I recommend having the whole manuscript revised again by an English native speaker to improve the "flow" of reading.